# Continuous Karakoram Glacier Anomaly and Its Response to Climate Change during 2000–2021

**Drolma Lhakpa** [1,2,3,4], **Yubin Fan** [1,2] **and Yu Cai** [1,2,*]

1  Jiangsu Provincial Key Laboratory of Geographic Information Science and Technology, Key Laboratory for Land Satellite Remote Sensing Applications of Ministry of Natural Resources, School of Geography and Ocean Science, Nanjing University, Nanjing 210023, China

2  Jiangsu Center for Collaborative Innovation in Novel Software Technology and Industrialization, Nanjing 210023, China

3  Tibet Institute of Plateau Atmospheric and Environmental Sciences, Tibet Meteorological Bureau, Lhasa 850000, China

4  Key Laboratory of Atmospheric Environment of Tibet Autonomous Region, Lhasa 850000, China

*  Correspondence: caiyu@nju.edu.cn

**Abstract:** Glacier mass balance is one of the most direct indicators reflecting corresponding climate change. In the context of global warming, most glaciers are melting and receding, which can have significant impacts on ecology, climate, and water resources. Thus, it is important to study glacier mass change, in order to assess and project its variations from past to future. Here, the Karakoram, one of the most concentrated glacierized areas in High-Mountain Asia (HMA), was selected as the study area. This study utilized SRTM-C DEM and ICESat-2 to investigate glacier mass change in the Karakoram, and its response to climatic and topographical factors during 2000–2021. The results of the data investigation showed that, overall, the "Karakoram Anomaly" still exists, with an annual averaged mass change rate of $0.02 \pm 0.09$ m w.e.yr$^{-1}$. In different sub-regions, it was found that the western and central Karakoram glaciers gained ice mass, while the eastern Karakoram glaciers lost ice mass in the past two decades. In addition, it was discovered that the increasing precipitation trend is leading to mass gains in the western and central Karakoram glaciers, whereas increasing temperature is causing ice mass loss in the eastern Karakoram glacier. Generally, decreasing net shortwave radiation and increasing cloud cover in the Karakoram restricts ice mass loss, while topographical shading and debris cover also have dominant impacts on glacier mass change.

**Keywords:** glacier mass balance; ICESat-2; SRTM; driving factors; Karakoram Mountains

## 1. Introduction

As essential climate variables (ECVs), glaciers are sensitive and reliable indicators of climate change, and they strongly contribute to a rise in sea level [1,2]. Glacier melt has always been a focus of attention over the cryosphere, and in this regard, glaciers located in High-Mountain Asia (HMA), as the highest plateau of the world, with an average elevation over 4000 m, could gain the attention of researchers. As greenhouse gas emissions are increasing and global warming is escalating in recent decades, the glaciers distributed in most regions in HMA, excluding the Karakoram and its adjacent regions (i.e., West Kunlun Mountains and East Pamir), are shrinking and melting due to the impact of climate change. This abnormal phenomenon linked to the Karakoram region is called the "Karakoram anomaly" [3–7]. Recent studies have confirmed that the Karakoram glacier's ice mass has been accumulating since the 1970s [8–10], which has made it a hot topic for glaciologists to address this paradox and has further led to proposals of more reliable tools and methods for glacier mass change forecasting in the Karakoram and HMA through different possible scenarios.

Mass balance is a direct and reliable indicator of glacier status [11]. Glacier change is very important in understanding climate fluctuations and the hydrological cycle in the mountains [12]. Mass balance is widely used to estimate glacier contribution to the runoff and sea-level changes through the mathematical relationship between the climate and glacier states. Monitoring changes in the glacier bodies and caps is a convenient method to recognize the rapid changes in the energy balance of the Earth's surface. IPCC AR6 has also reported that glaciers may display ice mass accumulation or recession because of internal dynamic mechanisms and local climate influences, such as the "Karakoram Anomaly," in the 21st century. Recent studies have shown that the "Karakoram Anomaly" may be connected to the low sensitivity of debris-covered glaciers in the Karakoram, to decreasing summer temperatures, or to increasing snowfall due to agricultural irrigation in this region [13].

Slight glacier mass gains and widespread surge activity are the two most prominent features of the Karakoram region [13]. The Karakoram glacial mass balance was −0.06 ± 0.19 m w.e.yr$^{-1}$ in 2000–2008 and 0.05 ± 0.19 m w.e.yr$^{-1}$ in 2008–2016, according to the AS-TER data; moreover, the overall glacial mass balance was −0.03 ± 0.07 m w.e.yr$^{-1}$ over the whole period of 2000–2016 [14]. This demonstrates that the Karakoram glaciers gained mass from 2008 to 2016. However, there has been a slight glacier mass loss for the Karakoram range in all of HMA since 2000. Over the Hunza Basin in Western Karakoram, the glacial extent decreased during 1992–1998, and then extended over 2008–2014 [15]. The Karakoram glaciers have experienced a nearly stable mass balance of −0.020 ± 0.064 m w.e.yr$^{-1}$ and −0.101 ± 0.058 m w.e.yr$^{-1}$ in the western and eastern parts, respectively, during 2000–2014 [16]. The glacial mass balance for different glacier types over the central Karakoram region analyzed for the time period 2000–2012 has shown the following results: surge-type glaciers −0.16 ± 0.11 m yr$^{-1}$, stable glaciers −0.08 ± 0.11 m yr$^{-1}$, advancing glaciers −0.13 ± 0.11 m yr$^{-1}$, and stable advancing glaciers −0.08 ± 0.10 m yr$^{-1}$, [17] where surge-type glaciers showed the most mass loss. The glacier elevation change during 2000–2012 was −0.19 ± 0.22 m yr$^{-1}$ in the Jammu and Kashmir glaciers located in the eastern region of the Karakoram, whereas the glacier elevation changes for surge-type and non-surge-type glaciers were −0.18 ± 0.22 m yr$^{-1}$ and 0.20 ± 0.22 m yr$^{-1}$, respectively [18]. In addition, the small and high-altitude glaciers demonstrated retreat, while the large and widely elevated glaciers were stable in the Karakoram Mountains [12]. Siachen Glacier exhibited near-stable conditions during 1986–2018, compared to the glaciers in other parts of the Himalayas [19]. Additionally, recent findings suggest that the Karakoram anomaly has extended to the nearby Western Kun Lun and Pamir Mountains [13]. In spite of a small discrepancy between the mass changes of glaciers at different time spans in the Karakoram, glacier accumulation trends were constant and were more dominant in the central Karakoram and non-surge type glaciers.

Over the past few decades, the positive glacier budgets in the Karakoram may have been caused by the increase in summer snowfall, and the decrease in net shortwave radiation. Most parts of the Karakoram produce high snow cover feedback [13]. Increasing and decreasing trends have been detected for cloud cover and incoming shortwave radiation over the Karakoram, respectively, using 36 years of WRF simulations [20]. Higher surface albedo and extended cloud cover could reduce the available net energy for snow and ice melting [13]. Previous studies indicated that glacier change is significantly correlated with summer temperatures, precipitation, surface wind speed, cloud cover, surface net radiation, and albedo [21]. Additionally, decreases and increases in summer temperatures and winter precipitation could be the main reasons for the "Karakoram Anomaly" and regional climate change could significantly impact the Karakoram glacier [22].

In previous studies, different RS productions and techniques, such as the TerraSAR/TanDEM, Advanced Spaceborne Thermal Emission and Reflection Radiometer (ASTER) [14,23,24], the Ice, Cloud, and Elevation Satellite-2 (ICESat-2), and the Spot5-HRS DEM [4,6,25], have been employed to evaluate the mass balance of the glaciers, which led to the distinct outputs and consequences reported in such studies. Due to the advantages

of ICESat-2 altimetry data in monitoring the thickness of ice and snow, ICESat-2 has been employed to monitor the mass balance of glaciers in the North and South Poles and mountainous glaciers [26,27]. Thus, it is significant to investigate the Karakoram glacier mass trend change in the future for regional and global climate change assessment, and for environmental and water resource protection. What is the latest trend of the "Karakoram Anomaly," and what are the latest changes in the climate trends behind it? To answer these questions, this study employed the latest ICESat-2 altimetry data and the Shuttle Radar Topography Mission C-band (SRTMc) digital elevation model (DEM) to detect changes in the mass balance of glaciers in the Karakoram region from 2000 to 2021. The glacier mass balance in western, central, and eastern Karakoram, and the glacier changes at different altitudes were analyzed. Furthermore, the relationships between the glacier mass balance and regional climate factors, including air temperature, precipitation, radiation and clouds, as well as topographical shading and supraglacial debris cover, were explored and discussed in this study.

## 2. Materials and Methods

### 2.1. Study Area

The Karakoram is the mountain range spanning the borders of Pakistan, India, and China; it also extends into Afghanistan and Tajikistan [13], with an average elevation of over 5500 m. According to RGI 6.0, it has a glacier area of 22843.07 km², where 25.71%, 60.27%, and 14.1% of the glacier areas are located in the Western, Central, and Eastern Karakoram, respectively (Figure 1).

The climate of the Karakoram Mountains is mostly semiarid and strongly continental, controlled mainly by westerlies and influenced by the Indian summer monsoon [13]. The location and altitude of the Karakoram is beneficial for glacier ice accumulation and stability [28].

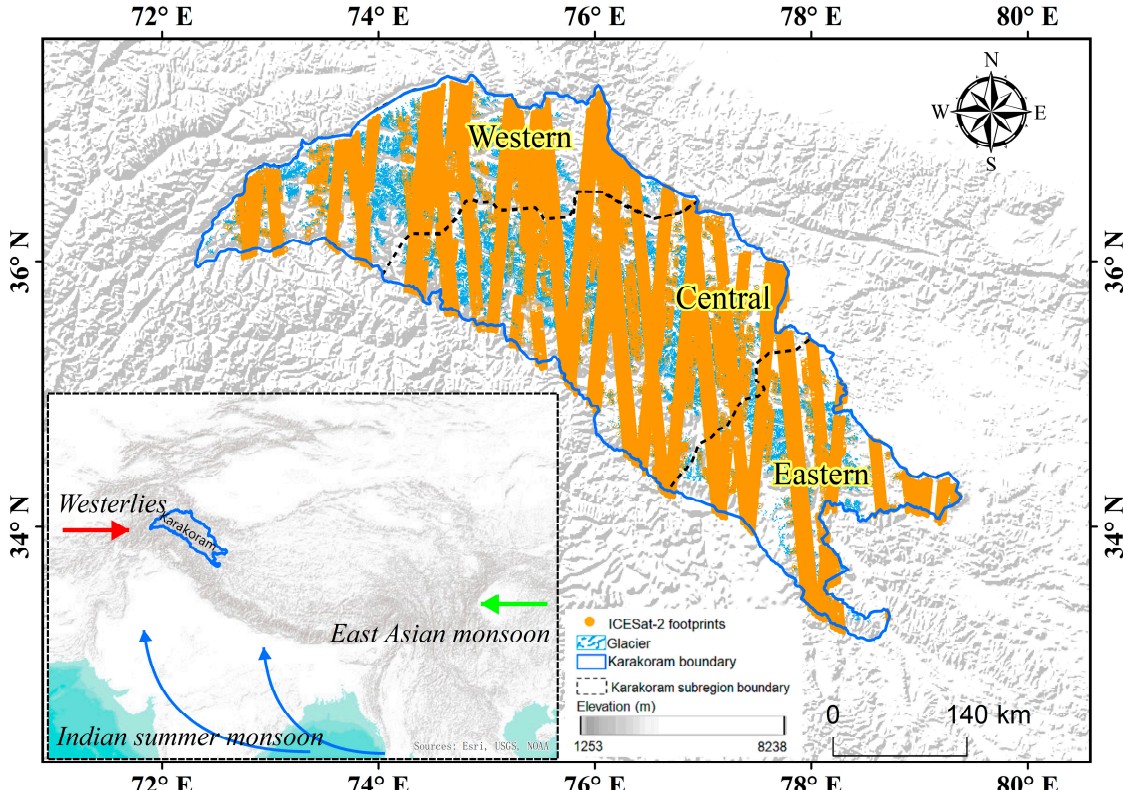

**Figure 1.** Glacial distribution in the Karakoram Mountains with ICESat-2 coverage. The glacier boundary is from the Randolph Glacier Inventory (RGI 6.0, http://www.glims.org/RGI/). The base map is from the GTOPO30 DEM.

*2.2. Data*

2.2.1. ICESat-2

ICESat-2 was launched on 15 September 2018, with an orbital height of 500 km and a revisit cycle of 91 days. It was equipped with the Advanced Terrain Laser Altimetry System (ATLAS), which splits the transmitted laser pulse into three pairs, and sets the beam-pair separation at 3.3 km across the track. Each pair contains one strong beam and one weak beam, and two beams within a pair are separated by 90 m [29]. The footprint of ICESat-2 is 17 m, with an along-track sampling interval of 0.7 m. This monitoring scheme can provide glacier elevation information in unprecedented detail, which is essential for the estimation of glacier elevation and mass change.

The ATLAS/ICESat-2 L3A Land Ice Height (ATL06) product was developed from global geolocated photon data (ATL03) to estimate the land ice height, which was determined after correction for instrument bias [30]. The ICESat-2 data used in this study were obtained from the National Snow and Ice Data Center (http://nsidc.org, accessed on 23 September 2021). For comparison with SRTM data, and to minimize the influence of seasonal snow cover on glacier elevation, ICESat-2 data ranging from January 15 to March 15 in 2019, 2020, and 2021 were used to estimate the glacier mass balance, which contained more than 500,000 on-glacier footprints.

2.2.2. SRTM Data

The C-band SRTM was generated by interferometry processing from image data collected in February 2000. This study selected the C-band SRTM 1 arc-second (approximately 30 m) v3 product (hereafter, SRTMc) to provide the glacier topography of 2000, and its voids were filled with other DEMs (e.g., ASTER Global Digital Elevation Model, GDEM). X-band SRTM (hereafter, SRTMx) was acquired simultaneously with SRTMc, but it shows an 'X' stripe-like coverage due to its smaller swath width. The SRTMc and SRTMx have mean errors of 4.31 ± 14.09 m and 9.03 ± 37.40 m and root mean square errors of 14.74 m and 38.47 m, respectively [31]. The C-band SRTM data were obtained from the Consultative Group on International Agricultural Research-Consortium for Spatial Information (CGIAR-CSI, http://srtm.csi.cgiar.org/, accessed on 23 September 2021), and the X-band SRTM were obtained from the German Aerospace Center (https://download.geoservice.dlr.de/SRTM_XSAR/, accessed on 23 September 2021).

These two DEM datasets have the same 30 m resolution; they have the same horizontal datum (WGS84 datum) but different vertical datum. For the EGM96 geoid height for the SRTMc and the WGS84 height for the SRTMx, SRTMc can cover all glaciated regions over the Karakoram, while SRTMx only covers 38.49% (Figure 2). Due to the shorter wavelength of X-band radar, its penetration depth into snow/ice is smaller than that of C-band; therefore, this study used SRTMx to estimate the penetration depth into snow/ice of the SRTMc.

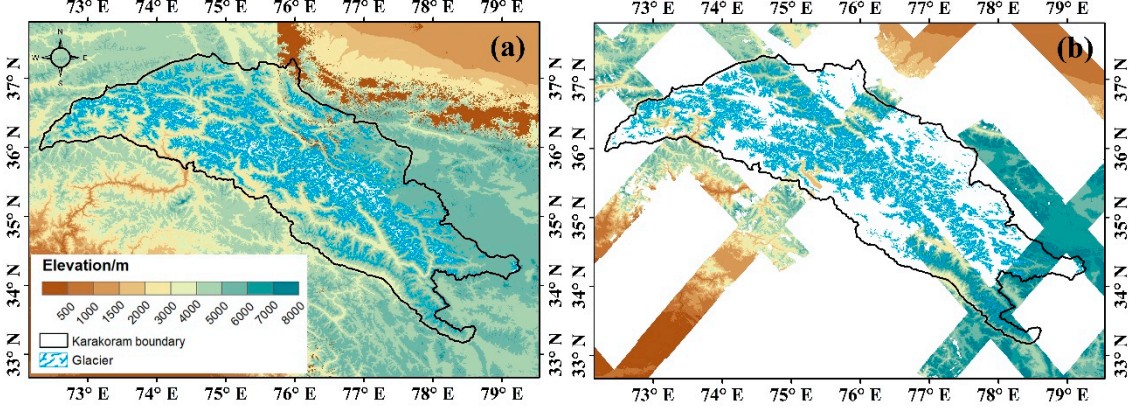

**Figure 2.** C-band (**a**) and X-band (**b**) SRTM data over the Karakoram Mountains.

2.2.3. ERA5 Reanalysis Data

Produced by the European Centre for Medium-Range Weather Forecasts (ECMWF), the ECMWF fifth-generation global atmospheric reanalysis dataset (ERA5) is the latest reanalysis dataset for global climate and weather research [32]. Current data includes data from 1950. The dataset was obtained from the European Centre for Medium-Range Weather Forecasts (https://cds.climate.copernicus.eu/, accessed on 31 December 2021).

This study selected 0.25-degree monthly products of 2 m temperature, total precipitation, mean surface net shortwave radiation flux, and total cloud cover from 2000 to 2021 to analyze the regional climate change trends and their connections with the glacier mass balance. In this research, to study seasonal climate trends, we considered June, July, and August to be the summer period, and December to the next year's February to be the winter period.

*2.3. DEM Coregistration*

Spatial offsets between different datasets were affected by differences in elevation and terrain aspect, and they could be corrected by the polynomial fitting trigonometric method of elevation change (*dh*) and aspect ($\psi$) in the off-glacier terrain regions [33], as expressed in Equation (1):

$$\frac{dh}{tan(\alpha)} = a * cos(b - \psi) + \frac{\overline{dh}}{tan(\overline{\alpha})} \tag{1}$$

where *dh* is the elevation difference between two datasets; $\psi$ and $\alpha$ represent the aspect and slope, respectively; $\overline{dh}$ and $\overline{\alpha}$ represent the average elevation difference and slope, respectively; and a and b show the length and angle of the offset, respectively.

First, the coordinate reference system of SRTMx was converted (from EGM96 to WGS84). To obtain higher accuracy of correction, each 1° × 1° SRTM grid was shifted to the ICESat-2 data over off-glacier regions via 10 iterations, as described in Equation (1). This study calculated the mean error (ME) in each 1° × 1° grid before and after co-registration, where the residual errors decreased greatly (Figure 3). Taking the grid with the highest improvement as an example (36–37°N, 80–81°E), the mean error decreased from 2.53 m to 0.04 m. About 77.1% of the grids had an offset length of less than ± 2 in both the X and Y directions, whereas for about 85.7% of grids, it was less than ± 3 m in the Z direction (vertical accuracy), as seen in Table 1.

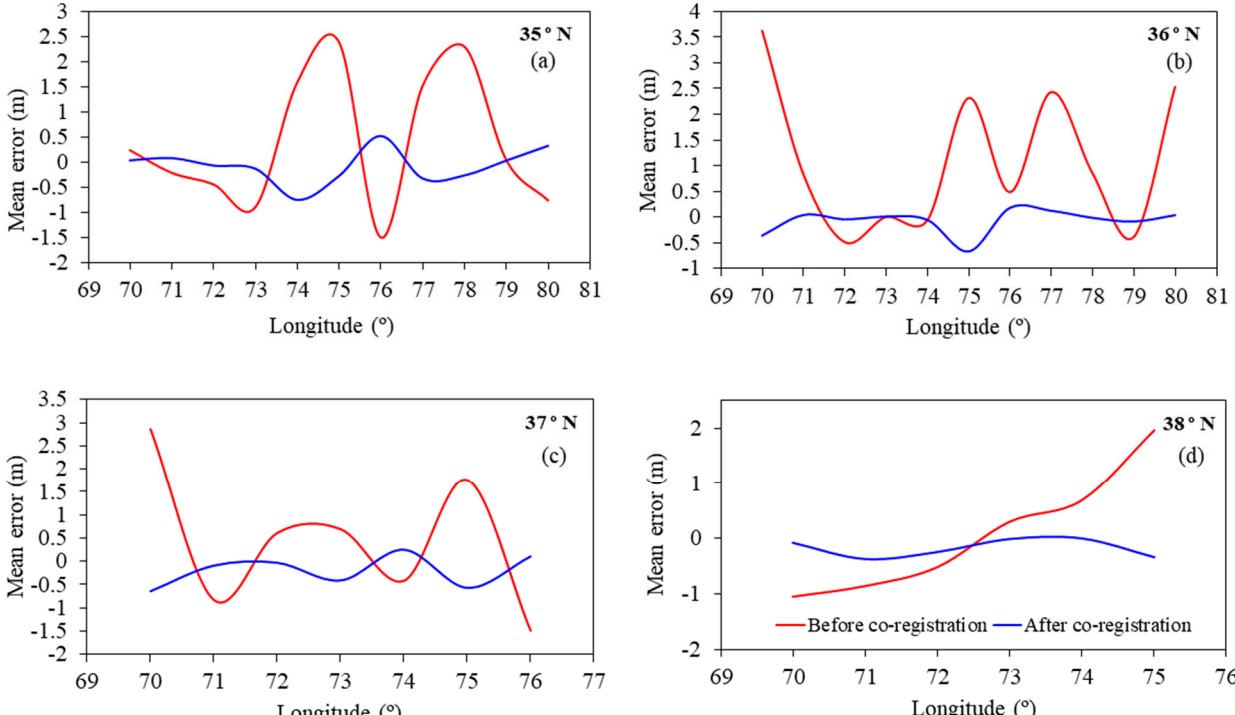

**Figure 3.** Mean error over 35°N (**a**), 36°N (**b**) 37°N (**c**), and 38°N (**d**) between ICESat-2 and SRTMc before (red line) and after coregistration (blue line).

**Table 1.** Aggregated displacement information during coregistration for each 1° × 1°grid over the Karakoram. (Note: positive values in the X and Y columns indicate offsets to the east and north, respectively, while positive values in the Z column indicate elevation increases.

| Coordinate | X (m) | Y (m) | Z (m) | Coordinate | X (m) | Y (m) | Z (m) |
|---|---|---|---|---|---|---|---|
| **N35E070** | 0.04 | 0.40 | −0.32 | **N36E077** | −1.75 | −2.89 | −2.76 |
| **N35E071** | −0.19 | −1.92 | 0.03 | **N36E078** | −3.51 | −1.70 | 0.46 |
| **N35E072** | 0.13 | −0.10 | 1.54 | **N36E079** | 3.42 | 4.31 | 2.49 |
| **N35E073** | −0.15 | 0.00 | 0.54 | **N36E080** | 0.00 | −2.75 | −0.59 |
| **N35E074** | 4.75 | 1.09 | 1.64 | **N37E070** | 0.24 | 1.69 | 2.94 |
| **N35E075** | −9.70 | −1.48 | 2.97 | **N37E071** | 1.47 | 0.66 | 2.87 |
| **N35E076** | −0.77 | 1.97 | 5.82 | **N37E072** | 0.74 | 0.44 | 1.00 |
| **N35E077** | −3.26 | −1.65 | −2.55 | **N37E073** | 6.56 | 1.47 | −1.40 |
| **N35E078** | −3.15 | 1.77 | 2.11 | **N37E074** | −0.06 | 1.22 | 2.25 |
| **N35E079** | 0.00 | 0.03 | −0.87 | **N37E075** | −2.85 | −3.30 | −2.48 |
| **N35E080** | 2.57 | 1.73 | −2.90 | **N37E076** | 6.99 | 2.57 | 5.27 |
| **N36E070** | 2.72 | −4.46 | 3.35 | **N38E070** | 1.97 | 0.90 | 4.15 |
| **N36E071** | 0.02 | −0.20 | −0.97 | **N38E071** | −0.01 | 0.80 | 2.42 |
| **N36E072** | −0.13 | −1.79 | 0.12 | **N38E072** | 0.21 | 0.48 | 1.06 |
| **N36E073** | 0.00 | 0.05 | 0.34 | **N38E073** | 1.68 | 1.90 | −0.46 |
| **N36E074** | 0.00 | 0.14 | 0.38 | **N38E074** | −2.30 | −1.62 | −0.73 |
| **N36E075** | −0.56 | −0.97 | 2.05 | **N38E075** | −7.60 | −4.78 | −4.17 |
| **N36E076** | −0.03 | −1.64 | −3.35 | | | | |

## 2.4. Penetration depth and Glacier Elevation Change

The penetration ability of the radar signal could lead to underestimation of the surface elevation of the glaciers [4]. Therefore, the radar penetration for the SRTMc into snow

and ice needed to be considered. However, penetration depth tends to be different for distinct surface types (i.e., elevation bands) [34]. Some studies have pointed out that the penetration depth of the C-band radar is twice that of the X-band radar [35,36]. Therefore, we corrected the penetration depth of the *SRTMc* by doubling the elevation difference between the *SRTMc* and *SRTMx* DEMs, as shown in Equation (2):

$$P = (SRTM_x - SRTM_c) \times 2 \tag{2}$$

where $P$ is the penetration depth of the *SRTMc*, and $SRTM_x$ and $SRTM_c$ represent the elevation values of the SRTMc and SRTMx data, respectively.

First, we re-projected the SRTMc and SRTMx data into the corresponding UTM projections at a resolution of 30 m, using cubic interpolation. Two DEMs were coregistered, as described by Equation (1). Then, calculated penetration depths and the depths exceeding ± 15 m were considered to be outliers [37]. After that, we calculated the average P for each 100 m elevation (Figure 4) and added this value to the SRTM*c* DEM. Regarding altitude bands with no efficient SRTMx DEM pixels for penetration depth correction, we chose their nearest neighboring elevation bands to fill the gaps.

The elevation changes (*dh*) in glaciers were corrected by Equation (3) in each 100 m bin, and footprints with *dh* that exceeded ± 200 m were eliminated. The measurement uncertainties (*h_li_sigma*) in the ATL06 product can represent the maximum error of the first photon bias correction and the linear fit error, which was used here for data initial quality control. Footprints containing *h_li_sigma* larger than 1 m were eliminated in further analyses.

$$dh = ICESat2 - SRTM_c - P \tag{3}$$

Here, *dh* is the elevation change, and *ICESat2* and *SRTMc* are the elevation measurements of both datasets. *P* is the estimated penetration depth.

### 2.5. Glacial Mass Balance Calculation

The mass balance was calculated with Equation (4), and an average glacier density ($\rho_{ice}$) of 850 ± 60 kg m$^{-3}$ was used to convert elevation change to mass, as described in [38]:

$$MB = \frac{\sum_i^n (dh \times S_i) \times \rho_{ice}}{S_{total} \times \rho_w} \tag{4}$$

where $n$ and $S_{total}$ represent the number of elevation bins and the total glacier area for a given glacier or region, respectively, and $\rho_w$ is the water density (1000 kg m$^{-3}$). $dh$ and $S_i$ are the mean elevation change and the glacier area within each 100 m elevation bin, respectively.

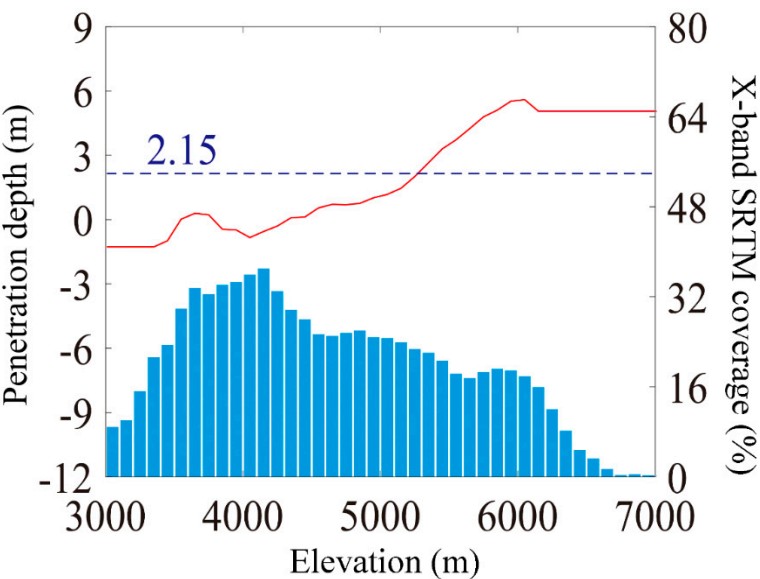

**Figure 4.** SRTMc penetration depth with elevation change. The solid red line represents the penetration depth at each elevation bin of 100 m, and the blue dotted line represents the mean penetration of each region (left *y*-axis). The bars display the SRTMx glacier coverage, which was calculated by dividing the SRTMx by the SRTMc glacier areas in each elevation bin (right *y*-axis).

*2.6. Data Uncertainty*

2.6.1. Elevation Change Uncertainty

Elevation change uncertainty includes the residual errors between ICESat-2 and SRTMc after coregistration, and the uncertainty of penetration calculations of SRTMc.

Glacial elevation change uncertainty ($U_{\Delta h}$) was calculated via Equation (5), after considering the average elevation change in the stable terrain combined with pixel resolution ($P_s$), autocorrelation distance ($D$), and a total number of measurements ($N_{total}$) [39,40]:

$$U_{\Delta h} = \sqrt{U_{se}^2 + U_p^2} \tag{5}$$

$$U_{SE} = \frac{\delta_{si}}{\sqrt{N_{eff}}} \tag{6}$$

$$N_{eff} = \frac{N_{total}\ P_S}{2D} \tag{7}$$

where $U_{SE}$ is the residual error, and $U_p$ is the penetration depth uncertainty of the SRTMc DEM, which was calculated from the standard deviation of the penetration depths per 100 m of elevation [37]. $\delta_{si}$ is the standard deviation of the elevation differences in the non-glacier regions. $P_S$ is the resolution of the ICESat-2 footprint (here, 20 m). $D$ is the distance of spatial autocorrelation; this study used the semi-variogram cloud to simulate the correlation distance of ICESat-2. The results showed that the decorrelation distance of ICESat-2 was approximately 2 km [41,42].

2.6.2. Mass Balance Uncertainty

Mass balance uncertainty was assessed with comprehensive approaches [40,42–45], and the final mass balance uncertainty ($U$) was related to four factors, including glacier elevation change uncertainty ($U_{\Delta h}$), glacier area uncertainty ($U_a$: ± 10%), penetration uncertainty ($U_p$: ± 0.12 m), and glacier density uncertainty ($U_d$: ± 60 kg m⁻³) [45], as shown in Equation (8):

$$U = \sqrt{U_{\triangle h}^2 + U_a^2 + U_p^2 + U_d^2} \tag{8}$$

## 3. Results

### 3.1. Penetration Depths

As shown in Figure 4, the penetration depth varied from −1.27 to 5.06 m with different elevations over the Karakoram, with a mean penetration depth of 2.15 m. The penetration depth increased with increasing elevation, and this kind of trend agrees with other studies for the Karakoram Mountains [16]. SRTMx DEM data did not cover the low elevation (2500–3400 m) and high elevation (6200–8600 m) areas, and the penetration values of these elevation bins were selected as −1.27 m and 5.06 m, respectively.

### 3.2. Glacial Elevation Change

The average glacier elevation change was $0.04 \pm 0.12$ m yr$^{-1}$ in the Karakoram Mountains from 2000 to 2021 (Figure 5). Generally, the change in the glacier elevation shifted from negative to positive as the elevation increased. A relatively stable elevation change could be seen in the elevation bins from 4000–6500 m, while the other bins exhibited instable glacier elevation changes, ranging from −1.32 m yr$^{-1}$ to 1.91 m yr$^{-1}$. Zero elevation changes occurred near the median elevation in West and Central Karakoram. The glaciers were distributed between 2750 m and 7550 m in West Karakoram, and the glacier of this sub-region showed an equilibrium state at 5250 m. The largest glacier elevation decrease of $-3.52 \pm 0.03$ m yr$^{-1}$ occurred at 2750 m, with the highest glacier elevation increase of $3.69 \pm 0.02$ m yr$^{-1}$ at 6650 m. Distributed between 2750 m and 7450 m, glaciers in Central Karakoram showed the same trend as those in West Karakoram, with the equilibrium line occurring at 5450 m. Despite being located at rather higher elevations, glaciers located in East Karakoram showed negative elevation changes across most elevation bins, with the largest elevation decrease of $-1.31 \pm 0.01$ m yr$^{-1}$ occurring at 4650 m.

Fluctuations in change rate and uncertainty could be seen in the higher elevation bins, which may have been contributed by the uneven distribution number of footprints. However, these higher elevation bins contained a small percentage of total glacier area, and therefore imposed less influence on the overall glacial mass balance estimation.

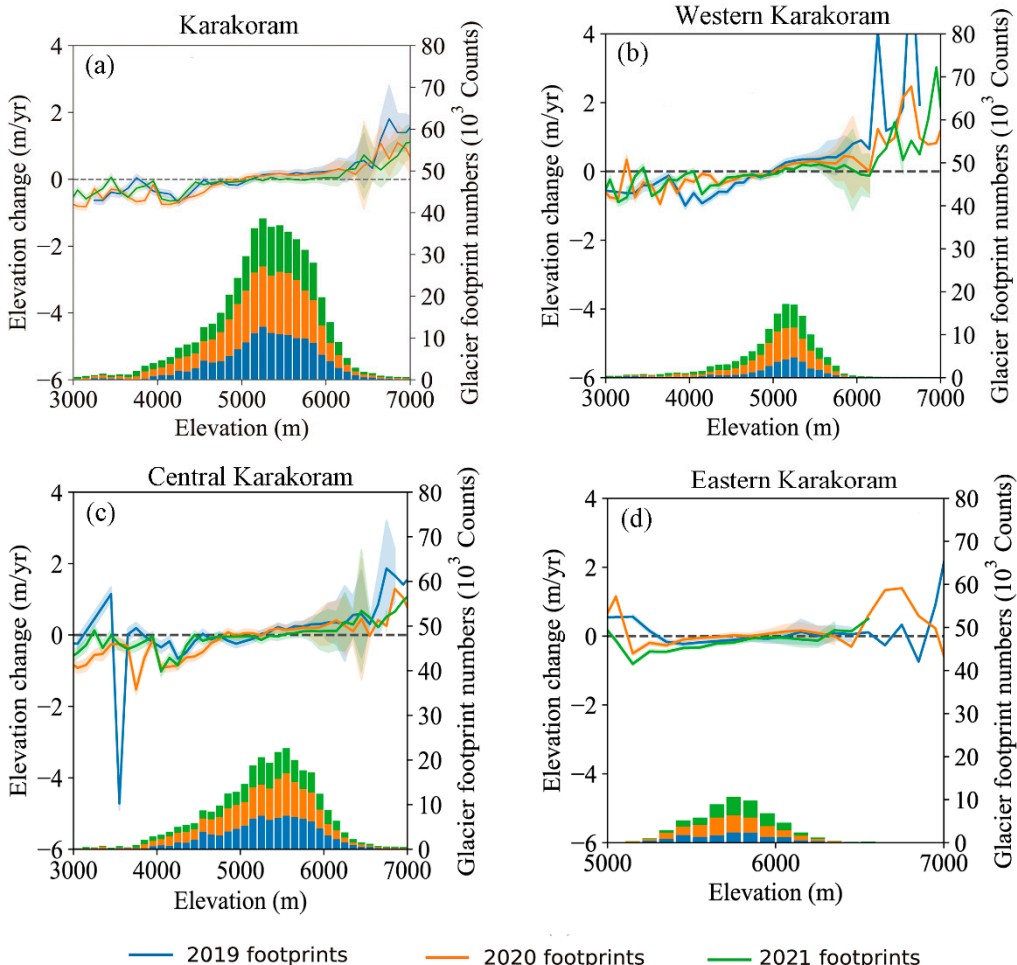

**Figure 5.** Altitudinal distribution of glacier elevation changes in the Karakoram. (**a**) Karakoram, (**b**) Western Karakoram, (**c**) Central Karakoram, and (**d**) Eastern Karakoram. The blue, orange, and green colors represent the elevation change rates calculated from footprints of 2019, 2020, and 2021 for each 100 m elevation band, respectively.

### 3.3. Glacial Mass Balance

The glaciers in the Karakoram Mountains were in a slight state of mass gain from 2000 to 2021, with a glacial mass balance of 0.02 ± 0.09 m w.e. yr⁻¹. The average glacial mass balances in Western, Central, and Eastern Karakoram were 0.04 ± 0.06 m w.e. yr⁻¹, 0.02 ± 0.08 m w.e. yr⁻¹ and −0.06 ± 0.04 m w.e. yr⁻¹, respectively, showing a clear mass decrease from west to east.

Glacial mass loss is one of the main reasons for rising sea levels in recent decades, and in this study, glacier mass changes in each basin in the Karakoram were calculated separately (Figure 6). The computed mass changes in all basins were less than ±1 Gt yr⁻¹. The western basin mostly exhibited mass loss, while the central and northern parts experienced mass gains, with the greatest mass loss and gain occurring in the Kharmong and Hotan basins, corresponding to glacial mass balances of −0.61 ± 0.14 Gt yr⁻¹ and 0.79 ± 0.43 Gt yr⁻¹, respectively.

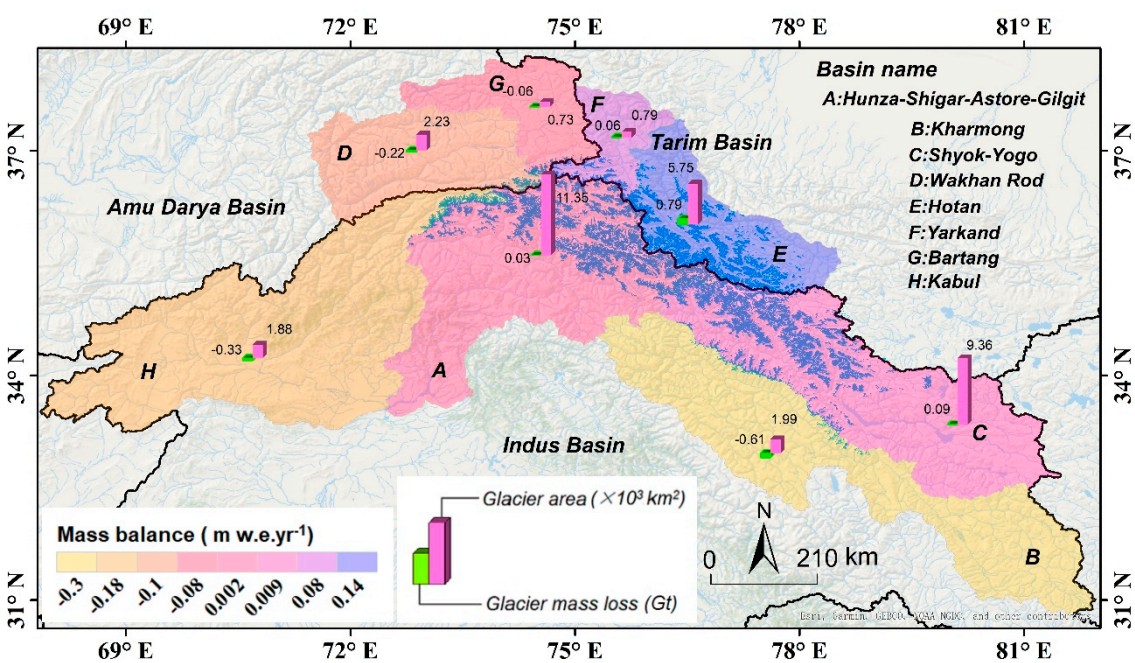

**Figure 6.** Glacial excess melt runoff for the major river basins in the Karakoram during 2000–2021. The color of the basin polygon represents the glacial mass balance, and the histograms represent the glacier area (pink) and glacier mass loss (green).

### 3.4. Changes in Climatic Factors

The annual average temperature in the Karakoram region showed an increase rate of 0.28 °C/10a from 2000 to 2021 (Figure 7a), among which the most pronounced increase of 0.44 °C/10a occurred in Western Karakoram, and the slowest increase of 0.09 °C/10a occurred in Eastern Karakoram. In contrast, the mean summer temperature showed different trends, where the entire Karakoram experienced a summer cooling trend of −0.06 °C/10a. The mean summer temperature was −1.28 °C in Central Karakorum over the past 22 years, whereas the temperature trend slope was as slow as 0.05 °C/10a (Figure 7b).

The total annual precipitation in the Karakoram region increased from 2000 to 2021 (2 mm/10a). The increasing trend of total annual precipitation was well clear (2.40 mm/10a) in Western Karakoram, followed by Eastern Karakoram, with a positive rate of 2 mm/10a (Figure 7c). The winter precipitation in the entire Karakoram showed a negative trend (−4.2 mm/10a), among which Western Karakoram experienced the most pronounced precipitation decrease of −5.1 mm/10a.

The annual averaged net shortwave radiation flux in the whole Karakoram region was decreased from 2000 to 2021 with a rate of −3.5 W m$^{-2}$/10a (Figure 7e). Similarly, other sub-regions of Western Karakoram, Central Karakoram, and Eastern Karakoram, were also experienced decreasing net shortwave radiation flux with the rates of 1.85/10a, 5.02/10a, and 2.12/10a, respectively. Similar to the annual averaged net shortwave radiation flux, summer-averaged shortwave radiation flux in the whole Karakoram showed a decrease rate of −3.4W m$^{-2}$/10a (Figure 7f). In different sub-regions, the trend of summer-averaged shortwave radiation was declining, with summer-averaged shortwave radiation flux values of 214.49W m$^{-2}$, 181.11W m$^{-2}$, and 244.64W m$^{-2}$, respectively. Furthermore, parts of Central, Western, and Eastern Karakoram showed the highest decrease rates of summer-averaged net shortwave radiation fluxes of 13.14/10a, 4.99/10a, and 5.36/10a, respectively.

The annual total cloud cover trend in the Karakoram increased from 2000 to 2021, with a rate of 0.02/10a (Figure 7g). Among all of the sub-regions, the cloud cover increase

rate of Western Karakoram was minimal (0.01/10a), while in parts of the central and eastern Karakoram, the increase rate was about 0.02/10a. Similarly, the Karakoram summer cloud cover increase rate was also detected to be 0.02/10a (Figure 7h).

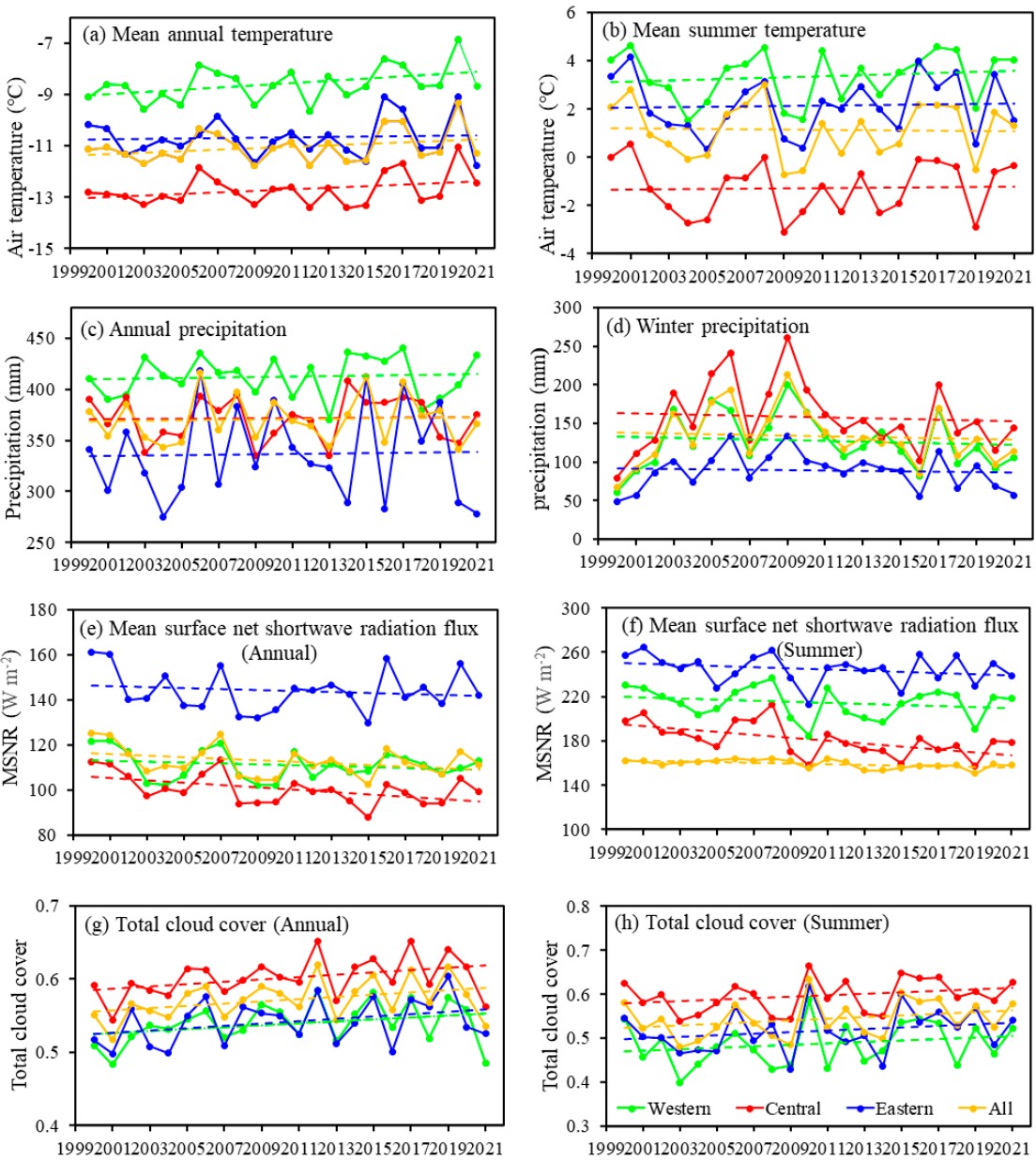

**Figure 7.** Change rates of (**a**) mean annual temperature, (**b**) mean summer temperature, (**c**) annual precipitation, and (**d**) winter precipitation; (**e**) net shortwave radiation flux, (**f**) summer net shortwave radiation flux, (**g**) total cloud cover, (**h**) summer total cloud cover, in the Karakoram from 2000 to 2021.

## 4. Discussion

### 4.1. Comparison with Previous Studies

The mass balance results obtained in this study based on ICESat-2 and SRTM are well compared with previous results for almost the same monitoring period, as seen in Table 2 [14,16,23,46,47].

Brun et al. [14] and Shean et al. [23] found that the entire Karakoram Mountain range experienced a slight mass loss, while the spatial pattern derived in this study was the same as that derived by SRTM [16,47]; thus, the mass balance shifted from positive to negative,

and from west to east. We attributed the reasons for these different patterns to the data sources used for the estimations. Brun et al. [14] and Shean et al. [23] mainly used DEMs derived from optical images that were free of the influence of penetration. In contrast, the penetration depth of SRTMc data needs to be considered; however, the difference in snow thickness, snow type, and ice density on the glacier surface may lead to scattering signal differences. It is rather difficult to quantify it without using in situ data, and although several approaches have already been proposed to address this issue [4,48,49], the penetration depth uncertainty still remains as the largest uncertainty source in glacial mass balance estimation, where the mass change differences between the SRTM and ASTER DEMs appear to be systematically biased.

In addition, differences in the study periods and regional divisions may lead to such discrepancies. Therefore, this study did not eliminate glaciers with areas less than 2 km², which are more sensitive to climate change [23]. This process scheme may make the mass balance estimation become slightly negative. However, we still captured the same spatial patterns as those found in previous studies.

**Table 2.** Comparison of glacial mass balance in different regions over the Karakoram with previous studies.

| Study Area | Study Period | Mass Balance (m w.e. yr$^{-1}$) | Data | Reference |
|---|---|---|---|---|
| Entire Karakoram | 2000–2021 | +0.02 ± 0.09 | SRTM, ICESat-2 | This study |
| Western Karakoram | 2000–2021 | +0.04 ± 0.06 | SRTM, ICESat-2 | This study |
| Central Karakoram | 2000–2021 | +0.02 ± 0.08 | SRTM, ICESat-2 | This study |
| Eastern Karakoram | 2000–2021 | −0.06 ± 0.04 | SRTM, ICESat-2 | This study |
| Entire Karakoram | 2000–2018 | −0.04 ± 0.04 | WordView/GeoEye DEMs, ASTER DEMs | [23] |
| Entire Karakoram | 2000–2021 | −0.03 ± 0.12 | NASADEM, ICESat-2 | [46] |
| Entire Karakoram | 2000–2016 | −0.03 ± 0.07 | ASTER | [14] |
| Western Karakoram | 2000–2014 | −0.02 ± 0.06 | TanDEM-X, SRTM | [16] |
| Central Karakoram | 2008–2016 | +0.12 ± 0.14 | SPOT, ASTER, SRTM | [47] |
| Eastern Karakoram | 2008–2016 | −0.24 ± 0.12 | SPOT, ASTER, SRTM | [47] |
| Eastern Karakoram | 2000–2014 | −0.10 ± 0.06 | TanDEM-X, SRTM | [16] |

*4.2. Climate Factors Influencing the Glacier Mass Balance*

Increasing trends in temperature could not only lead to glacier recession, but they could also be the main reason for the acceleration in glacier surface ablation and reduction in mass accumulation; consequently, increasing temperatures may be the reason for increasing the ice temperature, expanding glacier crevasse, breaking ice, and extension of the ablation zone. In addition, an increase in ice temperature could escalate ice percolation within the glacier accumulation zone and intensify ice accumulation. Oerlemans et al. [50] stated that to compensate ice mass loss due to a 1 °C increase in temperature, precipitation should be increased up to 25% (or even 35%). Generally, the annual mean temperature in the Karakoram is increasing, while the summer temperature is decreasing, as reported by some studies [2,51]. According to the negative correlation between temperature and glacier mass change, the Karakoram glacier mass gain during 2000–2021 could be mainly attributed to the summer temperature decline; however, glacier mass loss in Eastern Karakoram could be predominantly linked to the increasing trend in temperature.

In general, precipitation change is connected to altitudinal, seasonal, and regional variations. The westerlies bring a large amount of precipitation, rendering mass gains in the Karakoram glacier. Precipitation and its increasing rates are much greater than those recorded in meteorological stations [52]. Snowfall is the main source of Karakoram glacier accumulation, and with an increased altitude, total precipitation is increased [53]. Although precipitation type within the glacier area is mainly solid precipitation including

snowfall, by increasing the temperature, solid precipitation will be transformed into rainfall, mostly in the low-altitude glacier zone [54]. Generally, in the Karakoram, summer precipitation is increasing, and spring precipitation is decreasing [22]. Increasing summer snowfall can protect glaciers from melting, which is a solid reason for mass gains in the Karakoram glacier [13]. In addition, latent heat released by rainfall can hasten glacier ablation. To further explain the impact of precipitation on the Karakoram glaciers, this study investigated spatial variations in the annual mean precipitation trends in Karakoram glacierized regions during 2000–2021 (Figure 8a). The results showed that Western and Eastern Karakoram are the two most sufficient precipitation sub-regions, followed by Central Karakoram, which includes the largest glacier areas amongst all Karakoram sub-regions. However, winter precipitation is predominantly distributed over Central Karakoram, more so than that over western and eastern parts of the Karakoram from 2000 to 2021 (Figure 8b). Additionally, in the last two decades, summer temperature has been decreasing and annual precipitation has been increasing in Karakoram, which is beneficial for glacier ice accumulation, as also pointed out by Hewitt et al. [2].

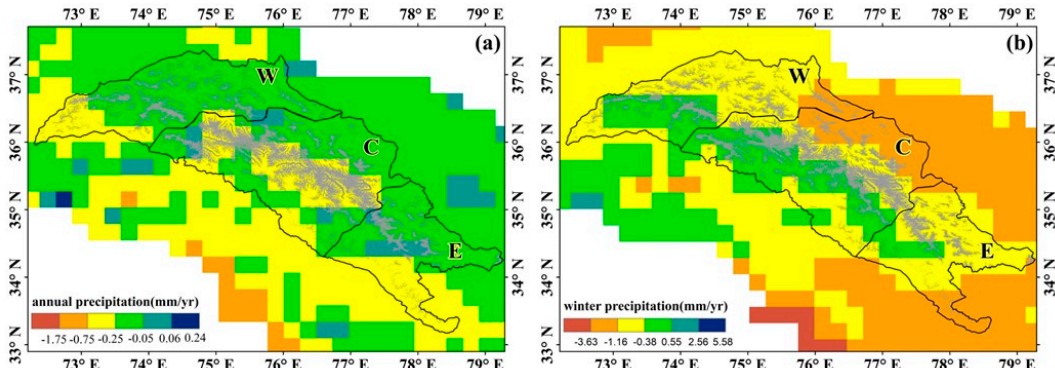

**Figure 8.** Spatial change map of annual (**a**) and winter (**b**) precipitation trends during 2000–2021. (W: Western, C: Central, E: Eastern).

Net shortwave radiation is the main driver for surface temperature variations in summer and autumn [55]. In the summer, increasing cloud cover, which impedes solar radiation, increases the humidity, and slows down the surface wind speed, all of which stop severe water loss due to evapotranspiration, and consequently decrease the glacier melting rate. Compared to other regions in HMA, net shortwave radiation is more significant for Karakoram glacier ablation. In the past ten years, summer snowfall has increased, and net shortwave radiation has decreased; such trends can be used to distinguish increasing trends in albedo in the Karakoram. By increasing the cloud cover and decreasing the air temperature, sensible heat is reduced due to the decrease in solar radiation absorption. Cloud cover increase can reduce the glacier melting rate [56]. From 2000 to 2021, except for cloud cover increasing, annual and summer mean net shortwave radiation have decreased, with a maximum decrease rate in Central Karakoram. Net shortwave radiation reduction retards glacier ablation, which can be used to explain glacier mass balance in the Karakoram, and to show the negative correlation between the glacier mass loss and net shortwave radiation. Farinotti et al. [13] stated that net shortwave radiation is the significant driver of Karakoram glacier ablation. Waqas and Athar [57] stated that increasing summer cloud cover can consequently slow down glacier ablation. In all such aspects, our research coincides with such studies.

According to the results of the current study, it was obvious that the summer mean temperature and precipitation showed decreasing and increasing trends over the Karakoram, respectively, from 2000 to 2021 (Table 3). Increasing the precipitation rate can not only contribute to glacier ice accumulation, but it can also reduce solar radiation over the region which further impedes glacier ablation and slows down the glacier recession rate. Generally, a decrease in temperature and an increase in precipitation could be considered

as the main reasons for glacier mass accumulation in the Karakoram region. Western Karakoram received the most precipitation, but it also experienced the highest rate of warming among the regions. Western Karakoram is the first region influenced by the westerlies, thus the water vapor brought from the westerlies is sufficient. Hence, an increase in temperature could not be enough to eliminate the recharge of precipitation to the region, resulting in a positive mass balance in Western Karakoram. The increasing rate of precipitation, lower temperature, and slower heating rate in summer play the most dominant roles for the positive mass balance in Central Karakoram. The winter precipitation helps to further glacier accumulation in the Central Karakoram as well. Conversely, the temperature dominates Eastern Karakoram, causing glaciers in this region to have a negative mass balance. Moreover, for the entire Karakoram, hampered glacier ablation from a decrease in net shortwave radiation and an increase in cloud cover are controlled by solar radiation.

**Table 3.** Meteorological trends in the Karakoram and its sub-regions during 2000–2021.

| | GMB | T(Y) | T(S) | P(Y) | P(W) | MSNR(Y) | MSNR(S) | TCC(Y) | TCC(S) |
|---|---|---|---|---|---|---|---|---|---|
| **Karakoram** | + | ↑ | ↓ | ↑ | ↓ | ↓ | ↓ | ↑ | ↑ |
| **WK** | + | ↑ | ↑ | ↑ | ↓ | ↓ | ↓ | ↑ | ↑ |
| **CK** | + | ↑ | ↑ | ↑ | ↓ | ↓ | ↓ | ↑ | ↑ |
| **EK** | − | ↑ | ↑ | ↑ | ↓ | ↓ | ↓ | ↑ | ↑ |

Note: WK: Western Karakoram, CK: Central Karakoram, EK: Eastern Karakoram. (a) GMB is glacier mass balance; (b) T (Y) is annual mean temperature trend; (c) T (S) is summer mean temperature trend; (d) P (Y) is annual precipitation trend; (e) P (W) is winter precipitation trend; (f) MSNR (Y) is annual mean surface net shortwave radiation flux trend; (g) MSNR (S) is summer mean surface net shortwave radiation flux trend; (h) TCC (Y) is annual total cloud cover trend; (i) TCC (S) is summer total cloud cover trend. "+" is glacier mass gain, "−" is glacier mass loss, " ↑ " is increasing, " ↓ " is decreasing.

*4.3. Topographical and Debris-Covered Glacier Impacts on Glacier Mass Balance*

Previous studies have confirmed that topographical shading can reduce ice ablation [58–61]. In the Karakoram, shading area over glacier surface can be up to 30.43%, especially in the low-altitude zone (9.5%), which is one of the most influenced sub-regions in HMA. Additionally, glaciers with thin debris cover (<5 cm) exhibited increasing ablation rates, compared to clean ice and thick debris-covered ice (>5 cm) [62–66].

Wang et al. [61] stated that most glaciers in the Karakoram are distributed in the northern aspect of Karakoram; in this area are also the least solar radiation-influenced areas, with 11.02% in shaded area. Moreover, there are approximately 20.82%, 20.45%, and 15.94% Western, Eastern, and Central Karakoram glaciers, respectively, distributed in the northern aspect of the Karakoram, which means that these glaciers are more influenced by topographical shading. Topographical shading protects glaciers from solar radiation, and this can explain glacier mass gain in the Karakoram.

Supraglacial debris cover is widespread in the Karakoram, where the total area of debris-covered glaciers is more than 16,800 km$^2$, of which 74.4% of the glaciers are larger than 1 km$^2$. From 1990 to 2020, the debris cover increased by 17.63 ± 1.44% (343.30 ± 27.95 km$^2$) [67]. High debris-covered glaciers are distributed over the Karakoram. This can reduce the melting rate of the glaciers to a certain extent. In addition, under the background of a westerly climate system, glaciers have sufficient water vapor supply sources to further develop. Therefore, Karakoram glaciers tend to be in a positive mass balance under the combined effect of debris coverage and climatic factors.

*4.4. Limitations and Future Research*

In spite of the higher altitude accuracy of ICESat-2, its limited spatial coverage over glacier surfaces has greatly restricted its application for glacial mass balance estimation. In some cases, ICESat-2 footprints only concentrate on glacier ablation or accumulation zones, which can alter the accuracy of mass balance estimation for whole glacier areas [3,46]. In this study, three years of ICESat-2 footprints were used together to obtain more reliable glacier mass change results, and this scheme may neglect the inter-annual glacier change among these years. Nonetheless, it is still reliable to apply ICESat-2 to large-scale glacier mass balance monitoring, for its small footprint size and high accuracy.

Generally, debris-covered glaciers according to the type of debris, have different ice mass changes compared to clean ice; this can alter glacier surface albedo, and may consequently alter the glacier mass balance [68–70]. Additionally, terrain factors such as aspect and altitude can also influence glacier mass changes [58,59,61]. Previous studies have stated that the debris-covered glaciers extensively distributed in the Karakoram have a great impact on the glacier mass change [4]. In addition, complicated topography is one of the reasons responsible for the occurrence and continuity of the "Karakoram Anomaly" [2,13]. Whether the Karakoram anomaly will continue in the future remains to be investigated with additional satellite data. Meteorological models and future climate forecasts can shed deeper light on the Karakoram anomaly. However, the ICESat-2 altimetry dataset has shown great advantages for glacier mass change estimation in mid-latitude regions. Therefore, it is expected that application of such altimetry datasets can improve glacier mass change detection.

## 5. Conclusions

Based on SRTMc DEM in 2000 and ICESat-2 altimetry data from January 15 to March 15 in 2019, 2020, and 2021, this study calculated the elevation difference and evaluated the mass balance of the Karakoram glaciers between 2000 and 2021. The main drivers of glacier change in the Karakoram region were discussed using ERA5 reanalysis air temperature, precipitation, cloud cover, and net shortwave radiation data. This study also investigated the topographic and debris-cover impacts on glacier mass change. The following remarks could be drawn, based on the obtained results:

(1). In general, the surface elevation of the glacier in low-altitude areas decreased, and the surface elevation of the glacier in high-altitude areas increased. During the 21 years, the annual mean change rate of the glacier elevation in the Karakoram region was $0.04 \pm 0.12$ m yr$^{-1}$ between 2000 and 2021, among which the change rate of glaciers distributed between 4000–6500 m a.s.l. was relatively small, while the glaciers at other altitudes experienced larger elevation changes.

(2). From 2000 to 2021, the Karakoram glaciers showed a slight positive mass budget of $0.02 \pm 0.09$ m w.e.yr$^{-1}$. However, the glacier mass balance was not uniform throughout the whole Karakoram region. The glaciers in Western and Central Karakoram experienced mass gains, while the glaciers in Eastern Karakoram experienced mass loss. During the 21 years, mass gains on the tops of the glaciers were the most evident.

(3). From 2000 to 2021, the annual mean air temperature and precipitation of the Karakoram Mountains increased. The annual warming trend gradually slowed from west to east, while the summer temperature in the Karakoram showed a decreasing trend. The Karakoram glaciers were sensitive to the air temperature and precipitation. The increasing precipitation was a main climatic driver for the glacier mass gains in Western and Central Karakoram, and increasing temperature contributed to the glacier mass loss in Eastern Karakoram. In addition, decreasing radiation and increasing cloud cover led to a reduction in received radiation by glaciers, which further inhibited glacier ice mass ablation.

(4). The topographic shadow and debris cover on the glacier surface increased the self-protection ability of the glaciers in the Karakoram, effectively restraining glacier

melting and playing a positive role in diminishing Karakoram glacier ice mass loss. Extraction of the quantitative relationships between influential factors (topographic shadow and debris cover) and glacier mass change could be considered and investigated in future research plans.

**Author Contributions:** D.L. wrote the manuscript and data process; Y.F. performed the ICESat2 and SRTM data process; Y.C. contributed to the conception of the study and supervised the research. All authors contributed to the discussion of the results, and to improvements in the manuscript. All authors have read and agreed to the published version of the manuscript.

**Funding:** This study was supported by the National Natural Science Foundation of China (grant No. 42061014) and the Second Tibetan Plateau Scientific Expedition and Research Program (STEP) (Grant No. 2019QZKK0201).

**Data Availability Statement:** The ICESat-2 data used in this study were obtained from the National Snow and Ice Data Center (http://nsidc.org, accessed on 23 September 2021). The C-band SRTM data used were obtained from the CGIAR-CSI (http://srtm.csi.cgiar.org/, accessed on 23 September 2021), and the X-band SRTM were obtained from the German Aerospace Center (https://download.geoservice.dlr.de/SRTM_XSAR/, accessed on 23 September 2021). The glacier boundaries were obtained from http://www.glims.org/RGI/rgi60_dl.html (accessed on 12 April 2021). The ERA5 data were from the European Centre for Medium-Range Weather Forecasts (https://cds.climate.copernicus.eu/, accessed on 17 November 2021).

**Acknowledgments:** We are grateful for funding from the National Natural Science Foundation of China and the Second Tibetan Plateau Scientific Expedition and Research Program. We are grateful to the satellite data providers: NSIDC for ICESat-2 and the German Aerospace Center for SRTM DEM product. The ERA5 reanalysis dataset was made available courtesy of ECMWF/Copernicus. We are also grateful to Chang-Qing Ke and Vahid Nourani for their numerous comments on this study. We also thank Jianwei Luo for his revision of this manuscript. Finally, we are grateful to the scientific editor, Luka Dzombic, and three anonymous reviewers, for their careful review of this manuscript, and for comments used to improve this manuscript.

**Conflicts of Interest:** The authors declare no conflicts of interest.

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
