# Peer review of "Continuous Karakoram Glacier Anomaly and Its Response to Climate Change during 2000–2021"

_remotesensing, doi:10.3390/rs14246281_

Round 1

Reviewer 1 Report

Comments on paper remotesensing-1967742

1. This paper presents a comprehensive study on the glacier anomalies Karakoram mountain in Northwest China.  The study serves as a good example for how to Judge Glacier Anomalies, and which is helpful for understanding the heterogenetic response mechanism of glaciers to global warming.  I believe this paper is a good one for the Remotesensing proceedings to publish.

2. However, the following comments should be useful to polish the quality of the presentation of the paper from readers’ benefit:

(1) I have found there are still many places where English grammar and presentation need to be corrected. the language still needs to be improved further. Please check the sentence carefully.

(2) The Introduction is not clear enough, I recommend rewrite abstract and introduction.

(3)  I believe the authors had collected many data sets for the research. But the data sources of different data are not clearly stated, so add the data sources clearly.

(4) One of the research aim of the paper is to discuss the air temperature and precipitation, but there is no any information about the temperature and precipitation data  in your paper data source partyou don't explain in your paper how you get the temperature and precipitation data, where they come fromPlease add these information. 

(5) In the discussion section you only discuss the differences between your research and previous research, but you did not mention the shortcomings and limitation of your paper, and the discussion section is too weak, please Rewrite the discussion section.

Author Response

Please see the attachment below.

Reviewer 2 Report

The study titled "Does the Karakoram Glacier Anomaly Persist? New Insight From ICESat-2" is an excellent effort to assess the Karakoram Glacier anomaly using ICESat-2 data. The authors do an excellent job analyzing the anomaly and use ERA-5 dataset to understand the reasons behind the variability. It can be accepted in the present form.

Author Response

Please see the attachment below.

Reviewer 3 Report

This contribution presents the works on calculating the glacier mass balance in the Karakoram during 2000 to 2020 via differencing ICESat-2 ATL06 product and SRTM-C DEM. The methods and results are basically right, and the English writing is good. However, before the publication of this contribution, there are some major issues need to be addressed.

Major comments:

(1) The methods are not innovative. The results on glacier mass balance are similar to that of previously published papers, even less accurate because of the effects of radar penetration over glacier surface. The monitoring period is only two years longer than previous measurement (2000-2020 vs. 2000-2018). The interpretation on the drivers of glacier change is also similar to previously published papers. The authors have already published the comprehensive and excellent work on calculating the glacier mass balance in High Mountain Asia during 2000 to 2020 via the same methods and dataset in last month (Fan et al., 2022). Moreover, Shen et al. (2022) published a paper on the 2003-2020 glacier mass balance in High Mountain Asia based on the same dataset. The monthly changes of glacier elevation in different regions of the HMA (including the Karakoram) from January of 2019 to December of 2020 were included in (Shen et al., 2022). Apparently, the results of glacier mass balance in the Karakoram based on ICESat-2 product and SRTM DEM have been included in (Fan et al., 2022) and (Shen et al., 2022). However, the main tone of this contribution is‘new insight from ICESat-2’. So, which knowledges are really new in this contribution?

The title of this contribution is attractive, but the present context does not match the title. The glacier mass changes and the drivers of glacier mass changes in the Karakoram are hot issues, and therefore have been thoroughly discussed. The question‘Does the Karakoram Glacier Anomaly Persist?’has already been answered by other papers. Although (Fan et al., 2022) and (Shen et al., 2022) did not highlight theKarakoram anomaly’, the results had already answered the question. Moreover, the results of Shen et al. (2022) should be more suitable to answer the question.

So, I would suggest the authors to focus on investigating other aspects of glacier dynamic in the Karakoram or on discussing the impacts of positive glacier mass balance. A much finer scale of investigation is also suitable, but the conclusions should be new or inspiring. Should the authors insist on focusing on the topic of ‘Karakoram anomaly’, you are suggested to modify the main tone of this contribution. It is a confirmation of ‘Karakoram anomaly persists’ through the ICESat-2 dataset, not ‘new’ insight. And the glacier mass balance results should be of the very recent years (say 2016-2021?), rather than the accumulative one over 2000-2020.

Fan, Y., Ke, C., Zhou, X., Shen, X., Yu, X., & Lhakpa, D. (2022). Glacier mass-balance estimates over High Mountain Asia from 2000 to 2021 based on ICESat-2 and NASADEM. Journal of Glaciology, 1-13. doi:10.1017/jog.2022.78 (online on 16 September 2022)

Shen, C.; Jia, L.; Ren, S. Inter- and Intra-Annual Glacier Elevation Change in High Mountain Asia Region Based on ICESat-1&2 Data Using Elevation-Aspect Bin Analysis Method. Remote Sens. 2022, 14, 1630. https://doi.org/10.3390/rs14071630

(2) The introduction should focus on the background and significance of this paper. You can review the status of glacier mass balance measurement in the Karakoram and the discussion of‘Karakoram anomaly’, and then conclude that the point that whether the Karakoram anomaly persists in recent years is not clear. Your discussion of climate change is not innovative. So, it is unnecessary to review the studies on the driver of ‘Karakoram anomaly’ in the Introduction. In addition, the significance of studying glacier mass balance in HMA and Karakoram should be very concise. It is a very common knowledge. Hence, the Introduction should be revised significantly.

(3) The discussion section is too short to make sense. The discussion should be significantly expanded.

Specific comments (no line number?)

(1) Remove‘Despite the negative mass budget of 0.33 ± 0.20 to 0.42 ± 0.20 m w.e yr-1 in the Himalayas, which had a negative mass budget’in the Introduction. It is unnecessary.

(2) Remove‘Glacial behaviour in the Karakoram is highly heterogeneous, both spatially and temporally, and its drivers are not yet fully understood’in the Introduction. You have no new insight of the drivers of glacier behavior.

(3) Rephrase‘However, thus far, little work with ICESat-2 has been done on glacier monitoring in the Karakoram region’in the Introduction. As mentioned above, two papers have reported the glacier monitoring in Karakoram using ICESat-2 dataset.

(4) ‘with a total area of approximately 22,800 km2‘ in the Introduction and ‘it has a glacier area of 22843.07 km2’ in the Study area are repetitive.

(5) Section 2.4:

Explain the‘measurement uncertainties (h_li_sigma)’

(6) Section 2.6:

How you determine the spatial autocorrelation distance of ICESat-2 ALT-06 product?

(7) Section 3.2:

The explanation of the heterogeneity of glacier elevation change rates in different altitude bin is unconvincing, and the logic of section 3.2 is very confusing. As the temperature rise and precipitation increase occur simultaneously, it is normal to see that the lower regions of glaciers become thinner while the higher regions become thicker. It is a result of usual ablation and accumulation, rather than the effects of debris cover and instability of precipitation pattern.

You say the glaciers in East Karakoram are a distribution at a rather low altitude. Seen from Fig.5, the altitude of glaciers in East Karakoram is much higher than that of West and Central Karakoram.

Remove‘also’ before ‘has snowfall in summer…’.

The expression of ‘The process of new snow formation’ is unprofessional.

Change ‘east-central’ into ‘east and central’.

You need to explain the remarkable fluctuations of glacier elevation changes rates in adjacent altitude bins very carefully. Almost all the glacierized regions in the northwest of HMA have more snowfall in summer than in winter, and the glaciers developed there are summer-accumulation type. You cannot simply attribute the remarkable fluctuations of glacier elevation changes rates in adjacent altitude bins to the summer snowfall in high regions. For example, the significant increase of glacier melting rate around 3500 m a.s.l. in Central Karakoram in 2019 looks like a gross error, and so do the large jumps of glacier accumulation rate around 6500 m a.s.l. in West Karakoram in 2019.

(8) The mass balance is calculated by multiplying the elevation change rate with glacier mass density (850 kg/m3 in this contribution). So, the expression‘the mean surface elevation difference and the overall mass balance of the Karakoram glaciers were 0.04 ± 0.12 m yr-1 and 0.02 ± 0.09 m w.e yr-1, respectively’in the abstract is strange. It is also unnecessary to list these two values together. Mass budget is sum of mass loss, and the unit of mass budget should be Gt yr-1, not m w.e yr-1. ‘m w.e yr-1’ is the unit of glacier mass balance. The authors cannot mix up these two concepts.

Author Response

Please see the attachment below.

Round 2

Reviewer 1 Report

Second Comments on paper remotesensing-1967742

1The author has made a lot of revisions to the paper, but the English still needs to be further improved.

2In this paper didn't add the limitations of this research, so I suggest you have to add the limitations of this research in the discussion section.

Author Response

Please see the attachment below.
